# Transarterial Radioembolization Planning and Treatment with Microspheres Containing Holmium-166: Determination of Renal and Intestinal Radionuclide Elimination, Effective Half-Life, and Regulatory Aspects

**DOI:** 10.3390/cancers15010068

**Published:** 2022-12-22

**Authors:** Christian Kühnel, Falk Gühne, Philipp Seifert, Robert Freudenberg, Martin Freesmeyer, Robert Drescher

**Affiliations:** 1Clinic of Nuclear Medicine, University Hospital Jena, Am Klinikum 1, 07747 Jena, Germany; 2Department of Nuclear Medicine, University Hospital Carl Gustav Carus, Technical University Dresden, 01307 Dresden, Germany

**Keywords:** transarterial radioembolization, TARE, holmium-166, radioactivity excretion, radioactive waste, effective half-life

## Abstract

**Simple Summary:**

^166^Ho-based transarterial radioembolization (TARE) procedures for liver cancer treatment can be safely applied in a hospital setting. It has been shown that a fraction of the injected radioactivity is excreted. Knowledge of the amount and nature of these excretions is vital for dosimetry, radiation protection, record keeping, and compliance with national and international regulations regarding waste disposal. Radioprotection measures should be taken, with particular attention to the urine of patients after treatment procedures. Regulations regarding radioactive waste must be considered.

**Abstract:**

After transarterial radioembolization (TARE) with microspheres loaded with holmium-166, radioactivity is excreted from the body. The aim of this study was to evaluate radioactive renal and intestinal excretions after TARE planning and treatment procedures with holmium-166-loaded microspheres and to correlate the findings with the intratherapeutic effective half-life. Urinary and intestinal excretions of patients who underwent TARE procedures were collected during postinterventional intervals of 24 h (TARE planning) and 48 h (TARE treatment). Whole-body effective half-life measurements were performed. Calibrations of the ^166^Ho measuring system showed evidence of long-living nuclides. For excretion determination, 22 TARE planning procedures and 29 TARE treatment procedures were evaluated. Mean/maximum total excretion proportions of the injected ^166^Ho were 0.0038%/0.0096% for TARE planning procedures and 0.0061%/0.0184% for TARE treatment procedures. The mean renal fractions of all measured excretions were 97.1% and 98.1%, respectively. Weak correlations were apparent between the injected and excreted activities (R^2^ planning/treatment: 0.11/0.32). Mean effective ^166^Ho half-lives of 24.03 h (planning) and 25.62 h (treatment) confirmed low excretions. Radioactive waste disposal regulations of selected jurisdictions can be met but must be reviewed before implementing this method into clinical practice. Inherent long-living nuclide impurities should be considered.

## 1. Introduction

For interventional locoregional treatment of patients with primary liver malignancies and liver metastases, transarterial radioembolization (TARE), also known as selective internal radiotherapy (SIRT), has become an established treatment modality [1]. For TARE, microspheres made of resin, glass, or poly-L-lactic acid (PLLA) are used [2]. Initially, TARE was performed only with yttrium-90 (^90^Y)-loaded microspheres. Since 2017, PLLA microspheres loaded with holmium-166 (^166^Ho) have been available for clinical use in Europe (QuiremSpheres^®^, Terumo, Leuven, Belgium) [3]. The half-life of ^166^Ho is shorter than that of ^90^Y (26.83 h and 64.1 h, respectively). The therapeutically active β-particles emit maximum energies of 1.77 MeV (abundance 49%) and 1.85 MeV (abundance 50%), respectively, which is also lower than that of ^90^Y (2.28 MeV; abundance 99.9%) [4]. ^166^Ho PLLA microspheres can be directly visualized on single photon emission computed tomography (SPECT) and magnetic resonance imaging (MRI) due to their gamma line assignment (80.6 keV; abundance 6.71%) and paramagnetic properties [5,6,7]. In contrast to TARE with ^90^Y, the same type of microsphere is used for planning and treatment procedures, which may increase the prediction of activity distribution in the liver [8].

Radioactive holmium is part of the polymeric matrix of biodegradable PLLA. Through emulsification and solvent evaporation, the holmium acetylacetonate crystals are processed into ^165^Ho acetylacetonate microspheres. Activation of ^166^Ho is then performed through neutron irradiation of the microspheres, which are sterilized simultaneously [9].

PLLA microspheres should settle in the capillary tissues and theoretically remain stable there, but it is known that small amounts of free radioactivity appear in the blood of treated patients and undergo renal and intestinal excretion [9,10]. Particularly for ^166^Ho-loaded microspheres, data are only available for small patient cohorts, and they do not take excretion after TARE planning procedures into account. For patient-derived radiation exposure, a dose rate between 8 and 37 µSv/h at a distance of 1 m, depending on the applied activity, has been reported [11]. The greater the amount of radioactivity excreted, the shorter the effective half-life compared to the radiological half-life. 

The aim of this study was to investigate urinary and fecal excretions of radioactivity in the first 24 and 48 h after ^166^Ho TARE planning and treatment, respectively. To estimate the overall biological elimination of the isotope, the intratherapeutic effective half-life of ^166^Ho was determined. 

## 2. Materials and Methods

This prospective, single-center study was approved by the institutional ethics committee (reg. no. 2019-1593) and registered in the German Clinical Trials Register (reg. no. DRKS00021427). All participating patients gave written informed consent. All cases were discussed by a multidisciplinary tumor board and recommended for TARE therapy. Treatment with ^166^Ho microspheres was performed with a uni- or bilobar approach, following the manufacturer’s instructions. 

The microspheres evaluated in this study are made of poly-L-lactic acid (PLLA), weigh 20 mg per million microspheres, and have a holmium content of 19–20% [12]. The specific activity of the microspheres used for TARE planning (“QuiremScout”, Terumo, Leuven, Belgium) is lower than that used for TARE treatment (“QuiremSpheres”, Terumo, Leuven, Belgium), with 4–5 MBq/mg and 12–15 MBq/mg microspheres, respectively. For TARE planning, two standard vials of 80 MBq and 170 MBq ^166^Ho are delivered, but up to three vials with personalized activities not exceeding a total of 250 MBq ^166^Ho are available. An activity of less than 10% of normal treatment activity is used to limit the therapeutic effects of the planning procedure. For TARE treatments, patient-specific ^166^Ho activities are used. The activity per treatment vial is determined by the number of microspheres [13]. Interventionary studies involving animals or humans and other studies that require ethical approval must list the authority that provided approval and the corresponding ethical approval code.

### 2.1. Measuring System Calibration

Activity measurements were performed in a shielded NaI scintillation detector (ISOMED 2100, Nuvia Instruments, Dresden, Germany) in 1000 mL Marinelli beakers (type 133 G-WMTJ, Nuvia Instruments, Dresden, Germany). For the calibration procedure of the given dose geometry to ^166^Ho, four reference activities from a sample of ^166^Ho chloride (1.138 MBq/L, 1.180 MBq/L, 1.949 MBq/L, and 1.990 MBq/L) provided by the manufacturer of the ^166^Ho PLLA microspheres (Quirem Medical B.V., AH Deventer, Netherlands) were measured daily with an energy window position at 81 keV and a window width of 25 keV over a period of 33 days (30.2 times the T_1/2_ of ^166^Ho). The remaining samples were evaluated for long-living nuclides in a high-purity germanium (HPGe) gamma-ray spectrometer for a measurement duration of 120 h (GC2018-CP5-PLUS-SL, Mirion Technologies (Canberra) GmbH, Rüsselsheim, Germany). 

### 2.2. Excretion Measurements

Excretion measurements were performed over 24 h after TARE planning (divided into two 12-hour collection intervals) and over 48 h after TARE treatment (four 12-hour collection intervals). Due to the varying amounts of excretions, the determination of activity concentration (Bq/mL) or absolute activity (Bq) in urine or feces was performed depending on the respective volumes: Samples with a volume of more than 1000 mL were measured in a Marinelli container, and the activity concentrations (Bq/L) were multiplied by the excreted volume. Samples with a volume of less than 1000 mL were diluted to 1000 mL, and the measured activity was corrected to an activity concentration with respect to the excreted volume. Fecal samples were homogenized to allow dilution. Radioactive decay was corrected to the middle of the respective collection intervals for urine and to the time of defecation for feces (absolute excretions, kBq) and used to calculate the ratio of excreted activity to injected activity (in kBq/GBq). To calculate the ratio of excreted to injected holmium (in %), decay correction to the time of injection was performed [10].

### 2.3. Effective Half-Life Measurements

To measure the intratherapeutic effective half-life of ^166^Ho, the dose rate arising from a patient was recorded at three time points per day (9 a.m., 1 p.m., and 8 p.m.) using dose rate monitors placed above their beds (DLMon with Geiger-Mueller counter tube, type 70004 SON16, STEP Sensortechnik und Elektronik Pockau GmbH, Pockau-Lengefeld, Germany). The probe SON16, with a measuring range of 1–500 μSv/h and an energy range of 35 keV–1.3 MeV, is equipped with a lead collimator on all sides that has aperture angles of +/− 30° mediolateral and +/− 60° mediocaudal/mediocranial. At the time of measurement, the patients were in a reproducible supine position. An average dose rate value was calculated over a period of 30 min with a fixed and reproducible setup (Figure 1).

### 2.4. Statistical Analysis

Statistical analyses were performed with the descriptive module using SPPS version 28.0 (IBM, Armonk, NY, USA) and with the Mann–Whitney U test for nominal or ordinal scaled parameters. Pearson’s correlation coefficients were calculated to assess potential relationships between injected and excreted activity.

## 3. Results

### 3.1. Measuring System Calibration

Results of the scintillation detector measurements over time showed the expected linear decay of ^166^Ho to ^166^Er down to a count rate of 150 counts per minute (cpm) (Figure 2). The calibration function calculated from these 116 readings was y = 0.0029x−15.6. After 150 cpm, no further decrease in the count rate was noted, suggesting the presence of long-living radionuclides. For further evaluation, the samples were transferred to the HPGe gamma-ray spectrometer. The resulting energy spectra confirmed several impurities, primarily the meta-stable ^166m^Ho (Figure 3). 

### 3.2. Patient and Procedural Characteristics

Twenty-three patients who underwent 22 TARE planning and 29 TARE treatment procedures with ^166^Ho PLLA microspheres were included in the study (Figure 4). Of these, six patients underwent seven TARE treatment procedures with ^166^Ho PLLA microspheres after planning was performed with ^99m^Tc-labeled human serum albumin microspheres (HSA B20, ROTOP Pharmaka GmbH, Dresden, Germany).

The clinical characteristics of the included patients are shown in Table 1. All procedures were technically successful and without complications. No relevant deterioration of liver or renal function occurred between the planning and treatment procedures, which were performed at intervals ranging from 9 to 16 days. Intra-individual differences in liver volumes were low and not statistically significant. The median injected activities for the TARE planning and TARE treatment procedures were 125 MBq and 3.5 GBq ^166^Ho, respectively.

### 3.3. Excretion

#### 3.3.1. Renal Excretion

After all 22 TARE planning and 29 TARE treatment procedures, radioactivity in the urine was measured. Urine volumes did not differ statistically significantly between collection intervals. For both procedure types, absolute excreted activity and proportion of injected activity were highest in the first collection interval (0–12 h) and decreased over the following collection intervals (Table 2, Figure 5A). As expected, absolute excreted activities were higher after TARE treatment than after TARE planning procedures due to the different injected activities, but the relative proportion of excreted activity from the injected activity was nearly equal between the two procedure groups (Table 2: proportions of injected activity [kBq/GBq and %]). The median ratios in the two collection intervals after TARE planning were 0.0019% (0–12 h) and 0.0013% (12–24 h). The median ratios after TARE treatment were 0.0021% (0–12 h), 0.0013% (12–24 h), 0.0014% (24–36 h), and 0.0010% (36–48 h) (Figure 5A).

After the TARE planning procedures, the mean excreted ^166^Ho activity in the first collection interval was more than double that in the second collection interval (2.8 kBq and 1.1 kBq, respectively). After the TARE treatment procedures, the mean excretion in the first collection interval was higher than in the remaining three collection intervals combined (71.5 kBq and 65.1 kBq, respectively) (Table 2: excreted activity [kBq]).

#### 3.3.2. Intestinal Excretion

During the patients’ hospital stays, defecations occurred after 12/22 (55%) TARE planning procedures and after 21/29 (72%) TARE treatment procedures (Table 3). No laxative measures were taken, since the study should reflect the normal course of patients. No clinical symptoms of obstipation were noted. The low number of two defecations in the collection interval 36–48 h (TARE treatment group) can be attributed to the relative immobilization of the patients on the ward. Fecal mass did not differ statistically significantly between collection intervals. 

Intestinal excretions were very low after all procedures (Table 3). Compared to renal excretions, intestinal excretions were delayed, with the highest absolute and proportional activities in the second (12–24 h, TARE planning) or third (24–36 h, TARE treatment) collection intervals (Figure 5B). Intestinal excretions were more variable over time.

#### 3.3.3. Total Excretion

Over periods of 24 h after the TARE planning procedures and 48 h after the TARE treatment procedures, the median proportions of 0.0038% and 0.0061% of the injected activity were excreted, respectively (Table 4). The majority of excretions were renal (TARE planning/treatment: mean 97.1% and 98.1%, respectively). After three TARE planning and two TARE treatment procedures in female patients, the measured renal proportions were considerably lower (89.4%, 82.5%, 87.9%, 90.1%, and 84.6%), while respective intestinal proportions were higher (10.6%, 17.5%, 12.1%, 9.9%, and 15.4%) than this average. Excluding these procedures, the ranges of renal proportions of total excreted activity after the TARE planning and treatment procedures were 94.9–100.0% and 95.4–100%, respectively.

For the TARE planning and treatment procedures, weak correlations were detected between injected and absolute excreted activity (R^2^ = 0.11 and 0.32, respectively; Figure 6). In none of the groups did the patient who received the highest or lowest injected activity also have the highest or lowest absolute excretion.

### 3.4. Effective Whole-Body Half-Life

Complete whole-body dose rate data were available for 17 TARE planning and 26 TARE treatment procedures. The mean effective ^166^Ho half-life after TARE planning was 24.03 ± 2.16 h (median: 24.6 h; range: 19.78–26.83 h). The mean effective ^166^Ho half-life after TARE treatment was 25.62 ± 0.99 h (median: 25.91 h; range: 24.12–26.83 h) (Figure 7).

## 4. Discussion

### 4.1. Excretion of Radioactivity after ^166^Ho TARE Procedures

This systematic evaluation of excretion and intratherapeutic effective half-life of patients who underwent TARE planning or TARE treatment showed that a small proportion of intra-arterially injected radioactivity was excreted. Compared to a previously published study, which evaluated only four patients after TARE treatment and reported an excretion proportion of 0.003 ± 0.002% (range: 0.001–0.005%) of the injected activity, no relevantly different results were observed in the current study [10]. After the TARE planning and TARE treatment procedures, radioactivity excretions of 0.0038 ± 0.0025% (range: 0.0004–0.0096%) and 0.006 ± 0.004% (range: 0.005–0.018%) of the injected activity occurred, respectively. 

In comparison to the previously mentioned study, the shortened collection intervals of 12 h for urine and feces in our study allowed for a more accurate assessment of the time course of excretions, showing that the highest urine activity excretions occurred during the first 12 h after the procedures, while fecal activity excretions were delayed (Figure 5). It can be hypothesized that if the majority of activity is secreted into the bowel (i.e., duodenum) with the bile, this delay reflects the time of passage from the duodenum to the rectum. Biliary secretion of activity has been described for ^90^Y TARE with resin microspheres [10]. Due to the limited overall collection intervals of 24 h and 48 h and the small number of patients with defecations, the maximum intestinal excretions may have occurred after the patients were discharged. Bakker et al. showed that inactive ^165^Ho may be excreted for up to 13 weeks after a TARE procedure. ^165^Ho and ^166^Ho levels in the blood and urine were initially low and declined quickly. A biphasic release of holmium from the microspheres has been described: a rapid initial phase and a delayed phase due to in vivo microsphere degradation [9]. The researchers also hypothesized that Ho^3+^, which is released from its acetylacetonate complex in the PLLA microspheres, forms complexes with albumin and phosphate in the bloodstream. These complexes may accumulate in the reticuloendothelial system (RES), which would mean that the amount of excreted ^166^Ho does not directly reflect the amount released from the injected microspheres.

A comparison of the TARE planning and treatment procedures in terms of renal excretions showed the higher absolute excretion that would be expected after TARE treatment procedures, but there were no statistically significant differences regarding the proportion of injected activity. For intestinal excretions, the excreted proportion of injected activity was statistically significantly lower after TARE treatment (only in the 12–24 h collection interval), but this was not reflected in total excretion values. No dependencies were found between excretions and tumor type, liver and renal function, or urine/feces volumes.

Overall, there was very good agreement with the reported ^166^Ho urine excretion of 0.003% (range: 0.000–0.022%) in 30 patients who received injected activities between 5.0 and 13.2 GBq [9]. Bakker et al. found significant correlations between injected activities and ^166^Ho content in the urine. In our study, absolute amounts of radioactivity excretions showed a weak positive correlation with injected activities, with patients excreting relatively low amounts despite high injected activities and vice versa (Figure 6). 

Previous data on intestinal excretions of ^166^Ho are available for only three patients whose intestinal excretions amounted to an average of 3.9% of the total excreted activities. In our cohort consisting of 12 (TARE planning) and 21 (TARE treatment) patients with defecations, the mean proportions were 2.9% and 1.8%, respectively. The proportion of intestinal excretions was above or equal to 10% after only four procedures (three planning and one treatment) in the three female patients included in the study. Based on our experience with the handling of excretions, contamination of feces by urine is assumed because not all patients were able to completely separate liquid and solid excretions.

Measurements of planning and intratherapeutic effective half-life confirmed marginal elimination of radioactivity from the body by biological means, with values only slightly shorter than or equal to the radiological half-life of ^166^Ho (26.83 h). Measurements over 24 h (TARE planning) showed a slightly shorter effective half-life than those over 48 h (TARE treatment), which is consistent with the higher excreted activity during the intervals 0–12/12–24 h compared with 12–24/24–48 h. However, the accuracy of the DLMon system, which measures the radiation arising from a patient lying in bed from a distance of 2 m, is limited for low radiation, explaining the more scattered data for the TARE planning procedures (Figure 5).

### 4.2. Long-Living Radioactive Impurities

As reported before, microspheres containing ^166^Ho also contain radioactive impurities, mainly ^166m^Ho, at approximately 130 Bq ^166m^Ho per 1 GBq ^166^Ho (1.3 × 10−5%) [9]. Experimental studies have shown that depending on the fabrication and composition method, up to 0.3 ± 0.1% of ^166^Ho is released into the buffer solution during the 24 h after production and may include additional impurities, including ^169^Yb (ytterbium), ^175^Yb, ^177^Lu (lutetium), ^140^La (lanthanum), ^143^Ce (cerium), and ^152^Eu [14,15,16]. Resin and glass microspheres loaded with ^90^Y may contain traces of ^88^Y, ^154^Eu, ^152^Eu, ^57^Co, and ^60^Co [17]. These impurities emerge during activation of the source material by neutron irradiation, not during radioactive decay in the patient. If it is assumed that the by-product ^166m^Ho is excreted to the same extent as ^166^Ho, a maximum total excretion of 14 Bq ^166m^Ho per injected 1 GBq of ^166^Ho PLLA microspheres would be expected. 

### 4.3. Regulatory Aspects of Radioactivity Release

Excretions of ^166^Ho (and ^166m^Ho) have to be taken into account when performing TARE in different countries with national regulatory requirements for the release of radioactivity into the public sewage system and the environment. The highest total excretion after a TARE treatment procedure in this study was 108 kBq per injected GBq ^166^Ho. If it is assumed that the by-product ^166m^Ho is excreted to the same extent as ^166^Ho, a maximum total excretion of 14 Bq ^166m^Ho per injected 1 GBq of ^166^Ho PLLA microspheres can be expected. The highest absolute excretion was 344 kBq ^166^Ho during 48 h (with an estimated 44.6 Bq ^166m^Ho). It occurred in an undiluted sample of 2585 mL of urine, resulting in concentrations at the time of excretion of 133 kBq/L ^166^Ho and 17.3 Bq/L ^166m^Ho. Among all patients, the highest ^166^Ho concentration in undiluted urine was 197.7 kBq/L (Table 2, TARE treatment, 0–12 h). In practice, this concentration will be diluted with a volume of 3, 6, or 9 L of water, adjustable at the filling valve per toilet flushing, and further diluted upon entering the sewage system. 

In the United States, the release of excretions from individuals undergoing therapy with radioactive material into the public sewage system is allowed [17]. The Canadian Nuclear Safety Commission published a limit of 1 MBq/L for ^166^Ho and 10 kBq/L for ^166m^Ho [18].

The current European Council directive suggests a ^166^Ho activity concentration limit for wastewater of 100 kBq/L [19]. This limit has been adopted in the United Kingdom [20]. In Germany, permissible activity concentration limits for wastewater discharging from radiation protection areas are 600 Bq/L for ^166^Ho and 200 Bq/L for ^166m^Ho, for up to 10,000 cbm of wastewater per year. A minimum stay of 48 h on a nuclear medicine ward with a dedicated sewage clearance system (decay plant) is required [21].

The Australian Radiation Protection and Nuclear Safety Agency has not set specific limits for waste containing ^166^Ho. This suggests that liquid radioactive waste from nuclear medicine departments may be disposed of in the sewage system, while solid waste should decay until disposed of [22].

In countries where local decay plants for radioactive wastewater are mandatory, such as Germany, ^166^Ho and ^166m^Ho are removed by biological systems or stored until the ^166^Ho concentration is below the permitted maximum concentration limits.

If the dilution of primary excretions in a toilet and sewage system is taken into account, our results show that TARE procedures with ^166^Ho-loaded microspheres would not be prohibited in countries with regulations for this nuclide and its primary by-product, ^166m^Ho. A sewage clearance or wastewater storage system may be necessary. In all circumstances, the relevant government agencies must be contacted before the treatment method is established in routine clinical practice. 

## 5. Conclusions

In summary, renal and intestinal excretion of ^166^Ho occurs at a low level, which was demonstrated via effective half-life determination. Radioactive excretions have to be considered to meet disposal regulations, which differ between jurisdictions. Exposure of staff to radiation from excretions can be minimized by avoiding contamination with the patient’s urine and by reducing the handling time of urine containers and bags.

## Figures and Tables

**Figure 1 cancers-15-00068-f001:**
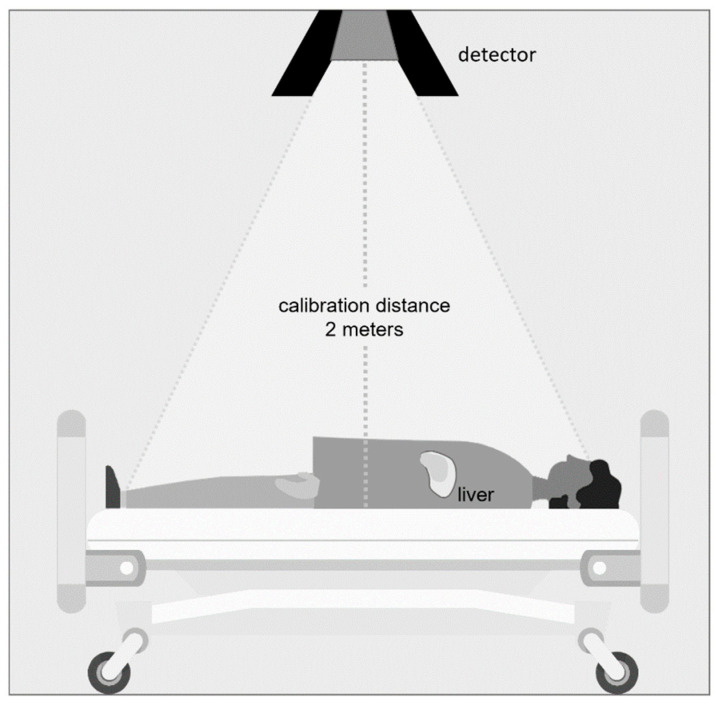
Setup of the whole-body dose rate meter.

**Figure 2 cancers-15-00068-f002:**
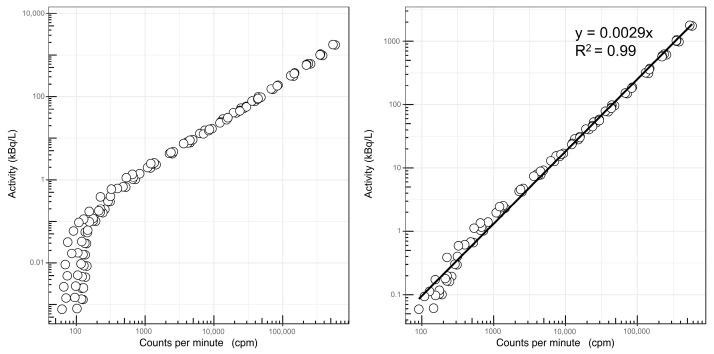
Measuring system calibration charts showing all measurements (circles), including long-living nuclides below 150 cpm (**left**) and without these nuclides (**right**).

**Figure 3 cancers-15-00068-f003:**
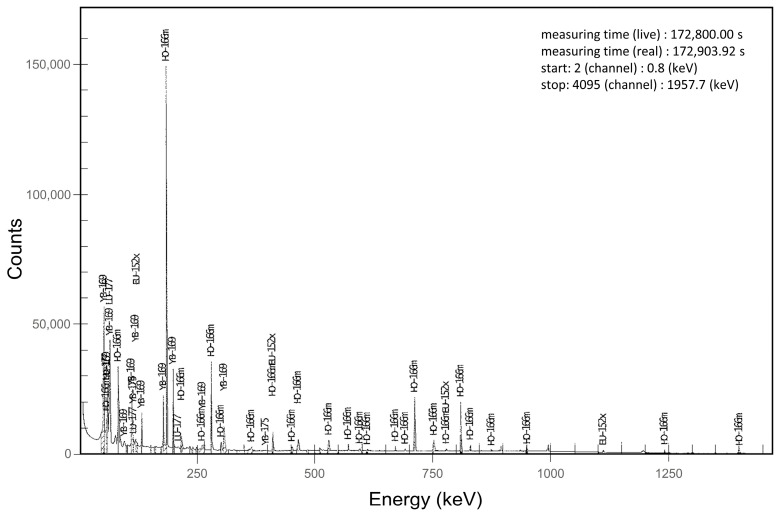
Energy spectrum of a 10 MBq sample of the ^166^Ho chloride calibration solution after 43 half-lives (48.1 d; residual ^166^Ho activity: 1.15 µBq). Trace amounts of ^166m^Ho (multiple peaks; estimated activity: 0.15 pBq), ^169^Yb, ^175^Yb, ^177^Lu, ^140^La, and ^143^Ce were detected.

**Figure 4 cancers-15-00068-f004:**
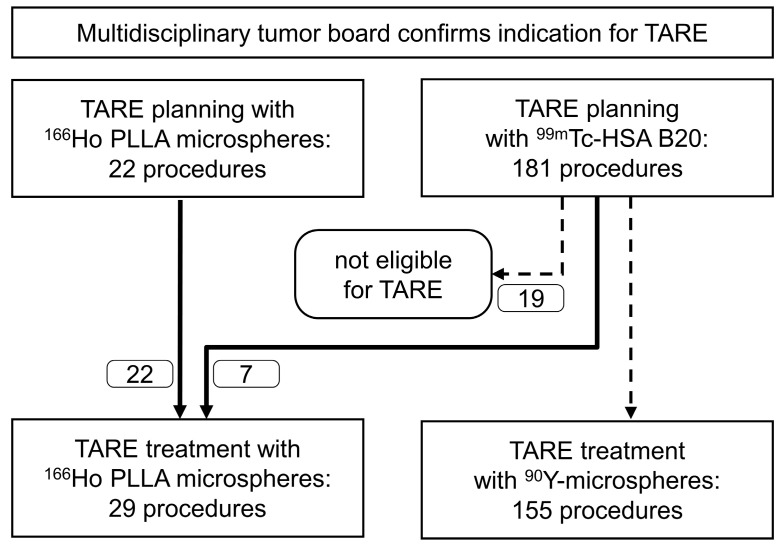
Over a period of 30 months, 203 TARE planning and 184 TARE treatment procedures were performed. All procedures involving ^166^Ho PLLA microspheres were consecutively included in the study.

**Figure 5 cancers-15-00068-f005:**
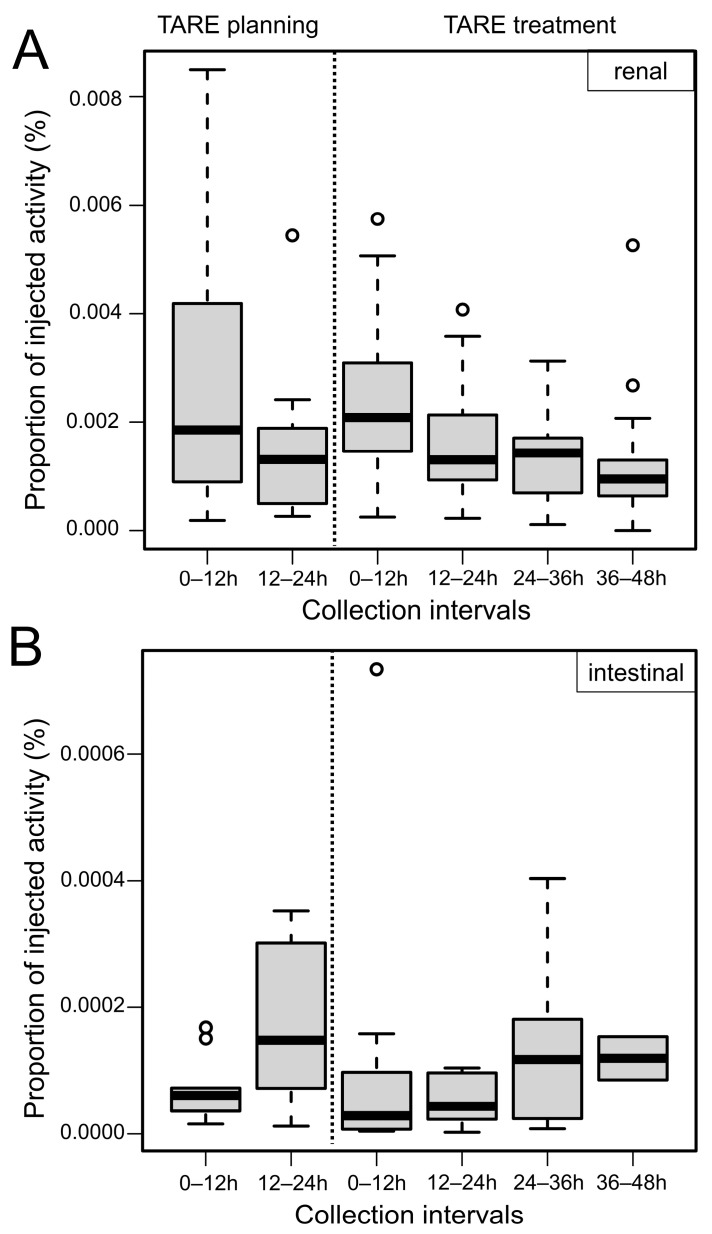
Relative renal (**A**) and intestinal (**B**) ^166^Ho excretion after TARE planning and treatment procedures.

**Figure 6 cancers-15-00068-f006:**
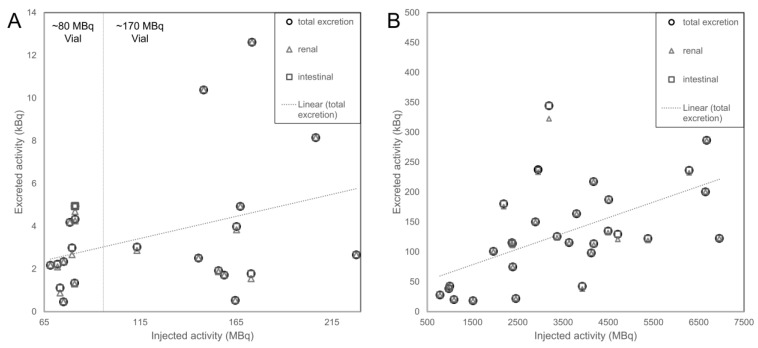
Correlation of absolute excreted to injected activity of ^166^Ho after TARE planning (**A**) and TARE treatment (**B**) procedures.

**Figure 7 cancers-15-00068-f007:**
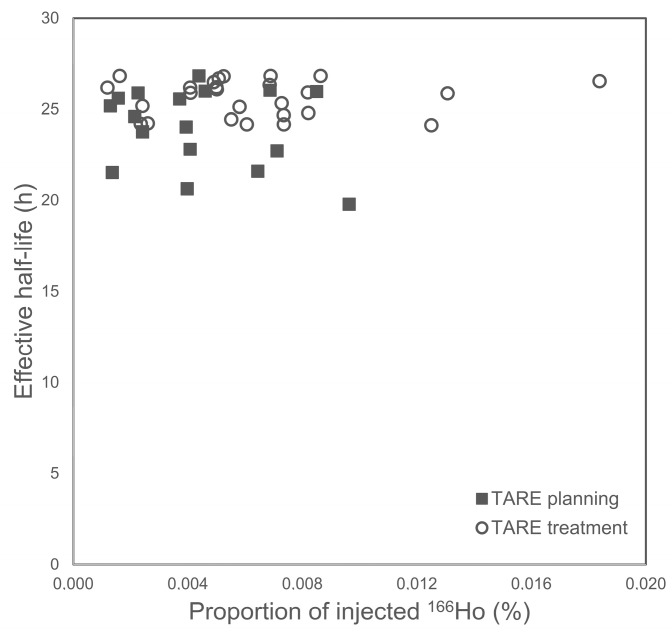
Correlation between effective half-life and excreted activity of ^166^Ho after TARE planning (24 h) and TARE treatment (48 h) procedures.

**Table 1 cancers-15-00068-t001:** Clinical and procedural characteristics.

	TARE Planning	TARE Treatment
No. of patients	17	23
Age (years) *	70 ± 9 72, 58–82	70 ± 8 68, 57–82
Sex	14 male, 3 female	20 male, 3 female
Tumor entity	12 HCC, 4 mCRC, 1 CCC	18 HCC, 4 mCRC, 1 CCC
Child–Pugh Score *	5.6 ± 1.1 5, 5–8	5.5 ± 0.9 5, 5–8
No. of procedures	22	29
eGFR (mL/min) *	77 ± 8.6 73, 43–100	81 ± 24.5 88, 40–105
Whole liver volume (mL) *	2242 ± 816.5 2265, 896–3731	2255.7 ± 644.5 2153, 1276–3843
Target liver volume (mL) *	974 ± 537.2 978, 190–1917	1022 ± 520.4 987, 208–2133
Injected activity (GBq) *	0.125 ± 0.051 0.129, 0.068–0.228	3.482 ± 1.771 3.368, 0.781–6.955

HCC, hepatocellular carcinoma; mCRC, metastatic colorectal cancer; CCC cholangiocellular carcinoma. * values are mean ± SD followed by median, range.

**Table 2 cancers-15-00068-t002:** Renal activity excretion.

	TARE Planning	TARE Treatment	*p*-Value
**0–12 h**			
Urinary volume (mL) *	929 ± 529.4 734, 162–2471	1021 ± 464.8 956, 244–2023	0.332
Excreted activity (kBq) *	2.8 ± 2.9 1.8, 0.2–12.6	71.5 ± 56 70.2, 3.2–286.6	<0.00001
Activity concentration (kBq/L) *	3.9 ± 4.3 1.8, 0.3–16.4	81.2 ± 56.3 70.1, 3.1–197.7	<0.00001
Proportion of injected activity (kBq/GBq) *	22.8 ± 19.5 15.9, 1.6–72.8	20.6 ± 11.8 17.9, 2.1–49.2	0.638
Proportion of injected ^166^Ho (%) *	0.0027 ± 0.0023 0.0019, 0.0002–0.0085	0.0024 ± 0.0014 0.0021, 0.0002–0.0057	0.624
**12–24 h**			
Urinary volume (mL) *	711 ± 419.5 591, 210–1843	760 ± 427.1 680, 134–1641	0.728
Excreted activity (kBq) *	1.1 ± 1.2 0.6, 0.2–5.1	33.1 ± 24.2 30, 3.5–89.2	<0.00001
Activity concentration (kBq/L) *	2.1 ± 2.6 1.2, 0.3–10.4	54.6 ± 50.4 46.8, 4–215.9	<0.00001
Proportion of injected activity (kBq/GBq) *	9.2 ± 7.9 8.2, 1.6–34.2	10.2 ± 6.3 8.2, 1.4–25.6	0.516
Proportion of injected ^166^Ho (%) *	0.0015 ± 0.0013 0.0013, 0.0003–0.0054	0.0016 ± 0.0010 0.0013, 0.0002–0.0041	0.441
**24–36 h**			
Urinary volume (mL) *		658 ± 348.2 588, 63–1235	
Excreted activity (kBq) *		18.3 ± 12.4 16.7, 2–45.9	
Activity concentration (kBq/L) *		31.2 ± 19.9 29.4, 5.2–76.5	
Proportion of injected activity (kBq/GBq) *		6.3 ± 4.1 6.6, 0.5–14.4	
Proportion of injected ^166^Ho (%) *		0.0014 ± 0.0009 0.0014, 0.0001–0.0031	
**36–48 h**			
Urinary volume (mL) *		779 ± 436.2 736, 59–1820	
Excreted activity (kBq) *		13.7 ± 11.9 10.5, 1.3–56.7	
Activity concentration (kBq/L) *		13.7 ± 12.2 10.5, 1.3–56.7	
Proportion of injected activity (kBq/GBq) *		4 ± 3.6 3.2, 0–17.8	
Proportion of injected ^166^Ho (%) *		0.0012 ± 0.0011 0.0010, 0.0000–0.0053	

* values are mean ± SD followed by median, range.

**Table 3 cancers-15-00068-t003:** Intestinal activity excretion.

	TARE Planning	TARE Treatment	*P*-Value
**0–12 h**			
Patients with defecation(s)	9	12	
Fecal mass (g) *	160 ± 188.7 114, 31–634	143 ± 90.1 104, 43–304	0.802
Excreted activity (kBq) *	0.07 ± 0.05 0.05, 0.01–0.15	2.5 ± 5.6 0.5, 0.1–20.1	0.00034
Proportion of injected activity (kBq/GBq) *	0.6 ± 0.5 0.5, 0.1–1.4	0.9 ± 1.8 0.2, 0.0–6.3	0.190
Proportion of injected ^166^Ho (%) *	0.0001 ± 0.0001 0.0001, 0.00000– 0.0002	0.0001 ± 0.0002 0.00003, 0.0000– 0.0007	0.190
**12–24 h**			
Patients with defecation(s)	8	6	
Fecal mass (g) *	129 ± 66.6 124, 55–242	113 ± 54.7 99, 64–221	0.749
Excreted activity (kBq) *	0.13 ± 0.08 0.12, 0.1–0.24	1.2 ± 1.1 1.0, 0.0–2.7	0.061
Proportion of injected activity (kBq/GBq) *	1.5 ± 1.1 1.3, 0.01–3.0	0.3 ± 0.3 0.3, 0.0–0.7	0.033
Proportion of injected ^166^Ho (%) *	0.0002 ± 0.0001 0.0001, 0.0000–0.0004	0.0001 ± 0.00004 0.00004, 0.0000–0.0001	0.081
**24–36 h**			
Patients with defecation(s)		11	
Fecal mass (g) *		100 ± 63.6 97, 28–257	
Excreted activity (kBq) *		2.3 ± 2.5 1.8, 0.2–8.8	
Proportion of injected activity (kBq/GBq) *		0.6 ± 0.6 0.5, 0.0–1.9	
Proportion of injected ^166^Ho (%) *		0.0001 ± 0.0001 0.0001, 0.0000–0.0004	
**36–48 h**			
Patients with defecation(s)		2	
Fecal mass (g) *		51 ± 41.7 51, 21–80	
Excreted activity (kBq) *		1.4 ± 0.2 1.4, 1.3–1.5	
Proportion of injected activity (kBq/GBq) *		0.4 ± 0.2 0.4, 0.3–0.5	
Proportion of injected ^166^Ho (%) *		0.0001 ± 0.00005 0.0001, 0.0001–0.0002	

* values are mean ± SD followed by median, range.

**Table 4 cancers-15-00068-t004:** Total activity excretion.

	TARE Planning	TARE Treatment
Time Frame	24 h	48 h
**Renal excretion**		
Excreted activity (kBq) *	3.6 ± 3.1 2.6, 0.5–14.6	128.2 ± 80.1 119.2, 18.3–322.6
Proportion of injected activity (kBq/GBq) *	30.2 ± 20.6 28.1, 3.2–72.8	38.5 ± 21.0 36.6, 8.9–101.1
Proportion of injected ^166^Ho (%) *	0.0037 ± 0.0025 0.0036, 0.0004–0.0096	0.0060 ± 0.0035 0.0051, 0.0012–0.0175
Proportion of total excreted activity (%) *	97.1 ± 4.8 100.0, 82.5–100.0	98.1 ± 3.3 99.5, 84.6–100.0
**Intestinal excretion**		
Excreted activity (kBq)*	0.2 ± 0.1 0.2, 0–0.3	3.3 ± 4.9 0.2, 0.1–21.8
Proportion of injected activity (kBq/GBq) *	0.7 ± 0.1 0.006, 0–0.3	0.7 ± 1.3 0.2, 0–6.8
Proportion of injected ^166^Ho (%) *	0.0002 ± 0.0001 0.0002, 0.0000–0.0005	0.0002 ± 0.0002 0.0001, 0.0000–0.0009
Proportion of total excreted activity (%) *	2.9 ± 4.8 0.0, 0.0–17.5	1.8 ± 3.3 0.5, 0.0–15.4
**Total**		
Excreted activity (kBq) *	3.6 ± 3.1 2.6, 0.5–12.6	130.5 ± 82.3 122.3, 18.3–344.4
Proportion of injected activity (kBq/GBq) *	30.2 ± 20.6 28.1, 3.2–72.8	39.2 ± 21.9 37.3, 8.9–108
Proportion of injected ^166^Ho (%) *	0.0038 ± 0.0025 0.0036, 0.0004–0.0096	0.0061 ± 0.0037 0.0053, 0.0012–0.0184

* values are mean ± SD followed by median, range.

## Data Availability

Not applicable.

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
