# Peer review of "Transarterial Radioembolization Planning and Treatment with Microspheres Containing Holmium-166: Determination of Renal and Intestinal Radionuclide Elimination, Effective Half-Life, and Regulatory Aspects"

_cancers, 2022, doi:10.3390/cancers15010068_

Round 1
Reviewer 1 Report
This paper gives an improvement on the potential fate of 116Ho microspheres after radioembolisation with microspheres. It is well written and could be quite useful in radiation protection. Some evaluations however need to be further explained as I somehow could not follow what was exactly done, see my specific points below.
I was wondering in what state 166Ho is excreted, presumably as chloride or were any analyses performed to check whether is was still bound to PLLA? When it is in its ionic form the ICRP models indicate 15% direct urinary excretion, 40% uptake in the liver, 40% in the skeleton and 5% in the spleen. Would this mean that in reality the release from the microsphere is a factor 6.6 (1/15%) higher, assuming that intra-arterial delivery will be equivalent. to intravenous. Most probably it is not an liver uptake is increased (not so bad) and renal excretion decreased. Please discuss this topic further.
Figure 2, page 4/5: It is unclear how the correction for the long-lived isotope admixture was performed, please explain more clearly. Also the curve indicated in part B is not the one shown. The Y=0.0029X-15.6 curve crosses the Y=0 axis at X=5483 cpm, please explain.
Table 1, page 6: Did any of these patients show extra-hepatic activity distribution on SPECT/CT imaging?
Table 2, page 7: To my knowledge a proportion of administered activity in kBq/GBq should correspond to a percentage which is a factor 10000 lower. 22.8 kBq/GBq would then correspond to 0.00228% but 0.0027% is indicated, please explain as in all further proportions I see comparable differences. Is the difference perhaps explained by the decay of 166Ho?
Figure 7, page 12: I find this graph very confusing, is there a relation between total excreted activity and effective half-life? Clearly a clearance half-life of much less than 27 h is needed to enable measurement of anything considering the low activities excreted, part B does not convince me there is a relation; I am sure the horizontal line at the mean value will be included in the 95% confidence interval.
Page14, line 303: The limit for Ho-166m is much lower than that for Ho-166, although unfortunately not included in the BSSD regulation. Could you indicate if it is possible to flush 166mHo without any problems?
Reviewer 2 Report
The authors present data on excretion of Holmium after treatment with Quiremspheres.
They provide data on the activity released, these data can be used by nuclear medicine department to discuss with regulatory agencies for the organization of treatment.
Main comments:
1- No clinical data on efficacy and safety is provided. I’m wondering whether this article is rightly targeted in an oncology journal, and should rather be proposed for nuclear medicine / imaging related journal.
2- The intestinal excretion was studied only for 48h, and the maximum excretion might not have been attained. This may require some additional discussion, even if this will probably not affect the overall results of low intestinal excretion as compared to urinary.
